# Tumor-infiltrating lymphocytes and macrophages as a significant prognostic factor in biliary tract cancer

**Ryota Tanaka**[1,2,3], **Shimpei Eguchi**[1], **Kenjiro Kimura**[1]*, **Go Ohira**[1], **Shogo Tanaka**[1], **Ryosuke Amano**[1], **Hiroaki Tanaka**[2], **Masakazu Yashiro**[2,4,5], **Masaichi Ohira**[2], **Shoji Kubo**[1]

**1** Department of Hepato-Biliary-Pancreatic Surgery, Osaka City University Graduate School of Medicine, Osaka, Japan, **2** Department of Gastroenterological Surgery, Osaka City University Graduate School of Medicine, Osaka, Japan, **3** Department of Medical Oncology, Sidney Kimmel Cancer Center, Thomas Jefferson University, Philadelphia, PA, United States of America, **4** Molecular Oncology and Therapeutics, Osaka City University Graduate School of Medicine, Osaka, Japan, **5** Cancer Center for Translational Research, Osaka City University Graduate School of Medicine, Osaka, Japan

* kenjiro@omu.ac.jp

## Abstract

### Background

The impact of tumor-infiltrating lymphocytes (TILs) and tumor-associated macrophages (TAMs) on the prognosis of biliary tract cancer (BTC) is not completely understood. Therefore, in our study, we investigated the effects of the various immune cells infiltration in tumor microenvironment (TME).

### Methods

A total of 130 patients with BTC who underwent surgical treatment at our institution were enrolled in this study. We retrospectively evaluated TILs and TAMs with immunohistochemical staining.

### Results

With CD8-high, CD4-high, FOXP3-high, and CD68-low in TME as one factor, we calculated Immunoscore according to the number of factors. The high Immunoscore group showed significantly superior overall survival (OS) and recurrence-free survival (RFS) than the low Immunoscore group (median OS, 60.8 vs. 26.4 months, $p = 0.001$; median RFS not reached vs. 17.2 months, $p < 0.001$). Also, high Immunoscore was an independent good prognostic factor for OS and RFS (hazards ratio 2.05 and 2.41 and $p = 0.01$ and $p = 0.001$, respectively).

### Conclusions

High Immunoscore group had significantly superior OS and RFS and was an independent good prognostic factor for OS and RFS.

**Data Availability Statement:** All relevant data are within the paper and its Supporting Information files.

**Funding:** The author(s) received no specific funding for this work.

**Competing interests:** The authors have declared that no competing interests exist.

## Introduction

The tumor microenvironment (TME) is composed of not only cancer cells but also an extracellular matrix and many types of non-cancerous cells, including fibroblasts, myeloid cells, and lymphocytes [1]. Immune cells, such as lymphocytes, neutrophils, monocytes, and dendritic cells, found in TME are called tumor-infiltrating immune cells (TIICs) [2]. Among TIICs, our institute has reported that tumor-infiltrating lymphocytes (TILs) and tumor-associated macrophages (TAMs) are involved in tumor progression and serve as prognostic factors in colorectal and breast cancer [3–5]. In recent years, TIICs are also known as a predictor of chemotherapy and immunotherapy effectiveness [6, 7].

Although TILs play a central role in anti-tumor immune response, in recent years, TAMs have become known as an important factor involved in tumor progression [8]. However, Zhang et al. revealed that high-infiltration of TAMs is associated with unfavorable prognosis in patients with gastric, urogenital, and head and neck cancer while it is associated with favorable prognosis in patients with colorectal cancer [9]. Their correlation with cancer prognosis remains unclear.

Biliary tract cancer (BTC) has an unfavorable prognosis. In recent years, not only surgical treatment and chemotherapy but also immunotherapy has been developed. However, these treatments for BTC are not satisfactory. Currently, immune checkpoint inhibitors have a confirmed efficacy for patients with BTC [10]. Although many researchers have reported TILs and TAMs as prognostic factors in BTC [11–16], we believe that it is necessary to consider TILs and TAMs together as cancer immunity in TME. The purpose of this study was to evaluate infiltration with lymphocytes and macrophages in BTC specimens that have undergone surgery in our department.

## Materials and methods

### Patient and tissue samples

Clinical data and formalin-fixed paraffin embedded (FFPE) tissues were obtained from 130 patients who underwent surgical treatment for BTC at our institution between 2001 and 2017 (Table 1). BTC includes intrahepatic, perihilar, and distal bile duct cancer, gallbladder cancer, and ampullary cancer. The surgical treatment for intrahepatic and perihilar cancer comprised partial hepatectomy and major hepatectomy with or without bile duct reconstruction. Gallbladder resection and extrahepatic bile duct resection with or without regional lymph nodes dissection were performed for gallbladder cancer. Pancreaticoduodenectomy was performed for distal bile duct cancer and ampullary cancer. None of the patients underwent preoperative radiotherapy or chemotherapy. Pathological findings were retrospectively evaluated following the Japanese classification of biliary tract cancers, third edition [17]. The TMN classification was reclassified following the American Joint Committee on Cancer system, eighth edition [18]. After surgery, the patients were followed up at every 3- to 6-month with tumor markers and enhanced computed tomography until 60 months. Recurrence-free survival (RFS) and overall survival (OS) are defined as the time from surgery to cancer recurrence or death. This study conforms to the Declaration of Helsinki and was approved by the Osaka City University Ethics Committee (approval number 924). Written informed consent was obtained from each patient.

### Tissue microarray construction

Tissue microarray (TMA) blocks with one 3.0-mm-diameter punch core per tumor were constructed from FFPE tissue blocks of resected specimens from primary site, as previously reported [19]. We ensured that representative tumor cell-rich areas are H&E-stained with a light microscope and were sent to create TMA blocks (S1 Fig).

**Table 1. Clinicopathological characteristics of 130 patients with BTC.**

| | | number |
|---|---|---|
| Sex | men | 71 |
| | women | 59 |
| Age, median (range) | | 68.5 (43–87) |
| Location of cancer | peripheral and distal bile duct | 56 |
| | intrahepatic bile duct | 20 |
| | gallbladder | 23 |
| | ampullary | 31 |
| T category | pT0 | 15 |
| | pT1 | 20 |
| | pT2 | 41 |
| | pT3 | 48 |
| | pT4 | 6 |
| Lymph node metastasis | absent | 88 |
| | present | 42 |
| Distant metastasis | absent | 121 |
| | present | 9 |
| Lymphatic invasion | absent | 51 |
| | present | 52 |
| Vascular invasion | absent | 86 |
| | present | 17 |
| UICC stage | 0 | 15 |
| | 1 | 22 |
| | 2 | 57 |
| | 3 | 34 |
| | 4 | 2 |
| Serum CEA level, ng/ml, median (range) | | 2.45 (0–86.5) |
| Serum CA19-9 level, U/ml, median (range) | | 30 (0–45152) |
| Chemotherapy | yes | 71 |
| | no | 59 |
| Recurrence | yes | 65 |
| | no | 65 |
| Outcome | death | 62 |
| | alive | 68 |
| Recurrence free survival, days, median (range) | | 544 (0–4160) |
| Overall survival, days, median (range) | | 786 (35–4157) |
| CD8 TILs, median (range) | | 40 (0–216) |
| CD4 TILs, median (range) | | 79 (0–330) |
| FOXP3 TILs, median (range) | | 21 (0–160) |
| CD68 TAMs, median (range) | | 92 (12–300) |

BTC: biliary tract cancer, UICC; Union for International Cancer Control, CEA; carcinoembryonic antigen, CA19-9; carbohydrate antigen 19–9, TILs; tumor infiltrating lymphocytes, TAMs; tumor associated macrophages.

## Immunohistochemical staining

TILs and TAMs were examined by immunohistochemical staining using BX50 DIC microscope (Olympus, Tokyo, JP). CD68 antibody was used as a pan-macrophage marker. Sections with a thickness of 4 μm were obtained from TMA blocks. Immunohistochemistry was done

as previously described [20]. Primary specific antibodies for CD4 (1:80 dilution; Dako, Glostrup, Denmark), CD8 (1:100 dilution; Dako, Glostrup, Denmark), FOXP3 (1:100 dilution; Abcam, Cambrige, UK), CD3 (1:100 dilution; Dako, Glostrup, Denmark), and CD68 (1:100 dilution; Leica Biosystems, Newcastle Upon Tyne, UK) were used.

### Evaluation of immunohistochemical staining

The immunohistochemical evaluation was performed by researchers independently. The number of TILs and TAMs around the tumor cells was evaluated with a microscope in three randomly selected fields at a magnification of × 400 and the average number was calculated (S2 Fig). All specimens were evaluated without any previous knowledge of the patients' clinical background.

### Determination of cutoff values

To set the cutoff values for the number of CD8+, CD4+, FOXP3+, and CD68+ cells, receiver operating characteristic (ROC) curve analyses for 5-year RFS were performed (S3 Fig). All patients were assigned into two groups, high-infiltration and low-infiltration groups, based on these cutoff values. The cutoff values were 40 for CD8+ cells, 48 for CD4+ cells, 29 for FOXP3 + cells, and 127 for CD68+ cells (Fig 1).

### Immunoscore in the tumor microenvironment

With CD8-high, CD4-high, and FOXP3-high TILs and CD68-low TAMs as one factor, the number of factors was counted in each case, and Immunoscore was assigned a score of 5 stages, from 0 to 4. Immunoscore was high for the score of 3–4 and low for the score of 0–2.

### Statistical analysis

Continuous variables were compared using the Mann-Whitney U-test. Categorical variables were compared using the chi-square or Fisher exact tests, as appropriate. Cox proportional hazard regression analyses were performed to identify prognostic predictors. The OS and RFS rates were estimated by the Kaplan–Meier method, and survival curves were compared using the log-rank test. Univariate and multivariate analyses were performed by cox regression

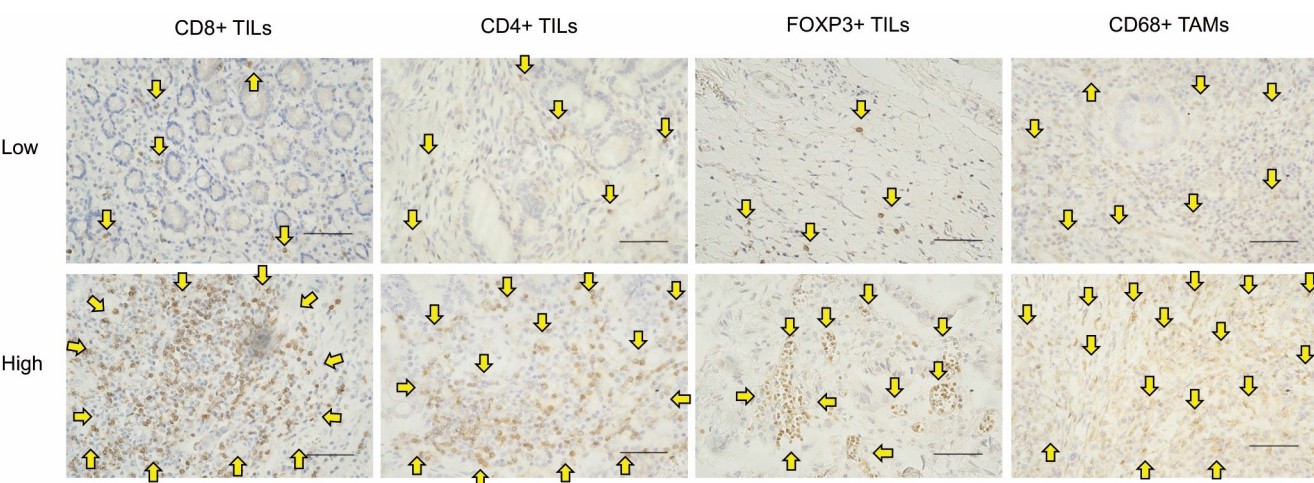

**Fig 1. Immunohistochemical staining.** Microscopic images showing high- and low-infiltration groups with CD8+, CD4+, FOXP3+, and CD68+ cells. Magnification is 400x, and the scale bar is 25 μm. Immune cells around the tumor are stained brown (arrows). Each patient is classified into the high- or low-infiltration group based on the cutoff value.

**Table 2. Correlation between clinicopathological features and TILs in 130 patients with BTC.**

| | | CD8 TILs | | P value | CD4 TILs | | P value | FOXP3 TILs | | P value |
|---|---|---|---|---|---|---|---|---|---|---|
| | | High | Low | | High | Low | | High | Low | |
| | | N = 72 | N = 58 | | N = 90 | N = 40 | | N = 51 | N = 79 | |
| Sex | men | 45 | 26 | *0.04 | 56 | 15 | *0.009 | 28 | 43 | 0.96 |
| | women | 27 | 32 | | 34 | 25 | | 23 | 36 | |
| Age, median (range) | | 70.5(51–87) | 67(43–86) | 0.43 | 69(43–83) | 68(43–87) | 0.91 | 67(43–84) | 69(46–87) | 0.19 |
| T category | pT0-2 | 40 | 36 | 0.45 | 53 | 23 | 0.88 | 33 | 43 | 0.24 |
| | pT3-4 | 32 | 22 | | 37 | 17 | | 18 | 36 | |
| Lymph node metastasis | absent | 51 | 37 | 0.39 | 64 | 24 | 0.21 | 36 | 52 | 0.57 |
| | present | 21 | 21 | | 26 | 16 | | 15 | 27 | |
| Distant metastasis | absent | 67 | 54 | 0.99 | 85 | 36 | 0.35 | 49 | 72 | 0.28 |
| | present | 5 | 4 | | 5 | 4 | | 2 | 7 | |
| Lymphatic invasion | absent | 32 | 19 | 0.6 | 39 | 12 | 0.3 | 23 | 28 | 0.62 |
| | present | 30 | 22 | | 35 | 17 | | 21 | 31 | |
| Vascular invasion | absent | 54 | 32 | 0.23 | 63 | 23 | 0.47 | 36 | 50 | 0.69 |
| | present | 8 | 9 | | 11 | 6 | | 8 | 9 | |
| UICC stage | ≦2 | 54 | 40 | 0.44 | 68 | 26 | 0.21 | 40 | 54 | 0.21 |
| | >2 | 18 | 18 | | 22 | 14 | | 11 | 25 | |
| Serum CEA level | <5 ng/ml | 61 | 40 | 0.14 | 76 | 25 | 0.12 | 43 | 58 | 0.97 |
| | ≧5 ng/ml | 8 | 11 | | 11 | 8 | | 8 | 11 | |
| Serum CA19-9 level | <37 U/ml | 37 | 29 | 0.74 | 49 | 17 | 0.43 | 27 | 39 | 0.82 |
| | ≧37 U/ml | 33 | 23 | | 38 | 18 | | 24 | 32 | |

*p < 0.05

TILs; tumor infiltrating lymphocytes, BTC; biliary tract cancers, UICC; Union for International Cancer Control, CEA; carcinoembryonic antigen, CA19-9; carbohydrate antigen 19–9.

hazard model. Groups were considered to be significantly different at $p < 0.05$. All tests were performed using JMP software.

## Results

### Evaluation of infiltrating immune cells and clinicopathological characteristics

Cases with the number of infiltrating CD8+ TILs that varied from 0 to 216/high power field (HPF) (median 40) (S2 Fig) and 55.4% (72 out of 130 cases) were assigned to the high-infiltration CD8+ TILs group. Cases with the number of infiltrating CD4+ TILs that varied from 0 to 330/HPF (median 79) and 69.2% (90 out of 130 cases) were assigned to the high-infiltration CD4+ TILs group. Cases with the number of infiltrating FOXP3+ TILs that varied from 0 to 160/HPF (median 21) and 39.2% (51 out of 130 cases) were assigned to the high-infiltration FOXP3+ TILs group. Cases with the number of infiltrating CD68+ TAMs that varied from 12 to 300/HPF (median 92) and 70% (91 out of 130 cases) were assigned to the low-infiltration CD68+ TAMs group. (Table 1). High CD8+ and CD4+ TILs were statistically significantly associated with gender ($p = 0.04$ and $p = 0.009$, respectively; Tables 2 and 3).

### Association between TILs and survival outcomes

In the entire 154-patient population, the high-infiltration CD8+ TILs group showed longer OS [20]. Due to the difference of the population in this study, the high-infiltration CD8+ TILs group

**Table 3. Correlation between clinicopathological features and TAMs in 130 patients with BTC.**

|  |  | High | Low |  |
|---|---|---|---|---|
|  |  | N = 39 | N = 91 |  |
| Sex | men | 25 | 46 | 0.15 |
|  | women | 14 | 45 |  |
| Age, median (range) |  | 72(43–83) | 67(43–87) | 0.19 |
| T category | pT0-2 | 19 | 57 | 0.14 |
|  | pT3-4 | 20 | 34 |  |
| Lymph node metastasis | absent | 24 | 64 | 0.32 |
|  | present | 15 | 27 |  |
| Distant metastasis | absent | 36 | 85 | 0.82 |
|  | present | 3 | 6 |  |
| Lymphatic invasion | absent | 13 | 38 | 0.11 |
|  | present | 21 | 31 |  |
| Vascular invasion | absent | 29 | 65 | 0.73 |
|  | present | 10 | 26 |  |
| UICC stage | ≦2 | 29 | 65 | 0.73 |
|  | >2 | 10 | 26 |  |
| Serum CEA level | <5 ng/ml | 31 | 70 | 0.59 |
|  | ≧5 ng/ml | 7 | 12 |  |
| Serum CA19-9 level | <37 U/ml | 16 | 50 | 0.07 |
|  | ≧37 U/ml | 22 | 34 |  |

TAMs; tumor associated macrophages, BTC: biliary tract cancers, UICC; Union for International Cancer Control, CEA; carcinoembryonic antigen, CA19-9; carbohydrate antigen 19–9.

showed the tendency of superior OS (median OS 51.3 months) compared to the low-infiltration CD8+ TILs group (median OS 34.5; $p = 0.09$; Fig 2A). Similarly, the high-infiltration CD8+ TILs group showed the tendency of superior RFS (median RFS 38.1 months) compared to the low-infiltration CD8+ TILs group (median RFS 18.7 months; $p = 0.06$; Fig 2B). Patients with high CD4+ TILs infiltration showed significantly superior OS and RFS than those with low CD4+ TILs (median OS 51.4 vs. 26.4 months, respectively, $p = 0.009$; median RFS 45.9 vs. 9.2 months, respectively, $p < 0.001$; Fig 2C and 2D). There is no significant difference in OS between patients with high FOXP3+ TILs infiltration (median OS 53.5 months) and those with low FOXP3+ TILs infiltration (median OS 33.9 months; $p = 0.11$; Fig 2E). Patients with high FOXP3+ TILs infiltration showed significantly superior RFS (median RFS not reached) compared to those with low FOXP3 + TILs infiltration (median RFS 20.8 months; $p = 0.02$; Fig 2F).

## Association between TAMs and survival outcome

In the total patient population, patients in the low-infiltration CD68+ TAMs group tended to have superior OS (median OS 53.5 months) compared to the high-infiltration CD68+ TAMs group (median OS 32.4 months; $p = 0.12$; Fig 3A). Similarly, patients with low CD68+ TAMs infiltration showed the tendency of superior RFS (median RFS: 45.9 months) compared to patients with high CD68+ TAMs infiltration (median RFS 20.8 months; $p = 0.21$; Fig 3B).

## Univariate and multivariate analyses of TILs and TAMs

Univariate and multivariate analyses of the patients were performed with clinicopathological predictors and infiltrating immune cells for OS and RFS using the cox regression model

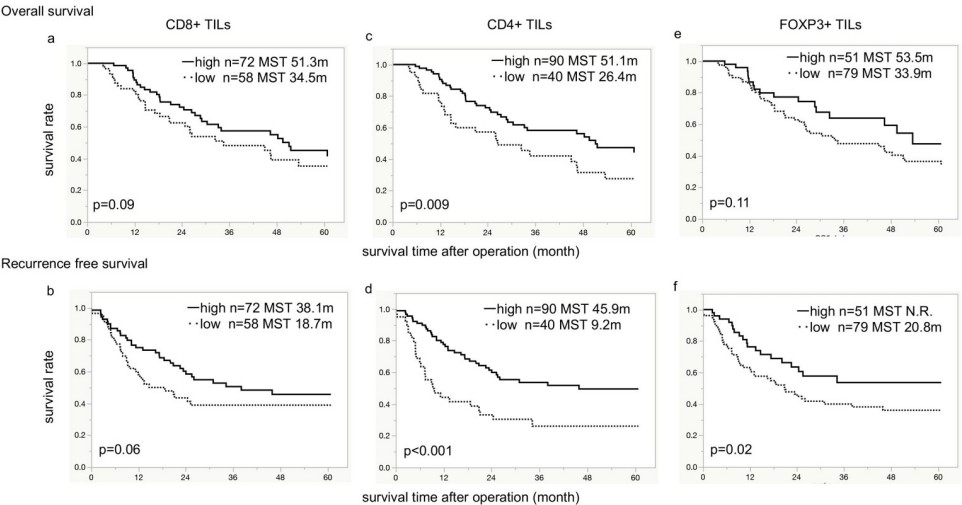

**Fig 2. Overall survival and recurrence-free survival for tumor-infiltrating T cells.** Kaplan-Meier survival curve indicates that the high-infiltrating CD4+ TILs group shows significantly superior OS and RFS than the low-infiltrating CD4+ TILs group. The high-infiltrating FOXP3+ TILs group shows significantly superior RFS than the low-infiltrating FOXP3+ TILs group. TILs: tumor-infiltrating lymphocytes, OS: overall survival, RFS: recurrence-free survival, MST: median survival time, N.R.: not reached.

(Tables 4 and 5). For OS, positive lymph node metastasis, presence of distant metastasis, and high serum CA19-9 level were independent poor prognostic factors (hazards ratio 2.0, 3.34, and 17.2, respectively; $p = 0.02$, $p = 0.007$, and $p = 0.04$, respectively; Table 4). For RFS, T classification (pT3-4) and presence of distant metastasis were independent poor prognostic factors with a hazards ratio of 2.20 ($p = 0.008$) and 3.1 ($p = 0.01$), respectively (Table 5). Although CD4+ and FOXP3+ TILs were a significantly favorable prognostic factor by the univariate analysis, the multivariate analysis did not show statistical significance for TILs. Furthermore, even if CD8+, CD4+, and FOXP3+ TILs were scored same as Immunoscore (TILs score; 0–3), TILs score was not an independent prognostic factor for OS and PFS (a hazards ration of 0.59, $p = 0,053$ and 0.71, $p = 0.22$, respectively: S1 Table).

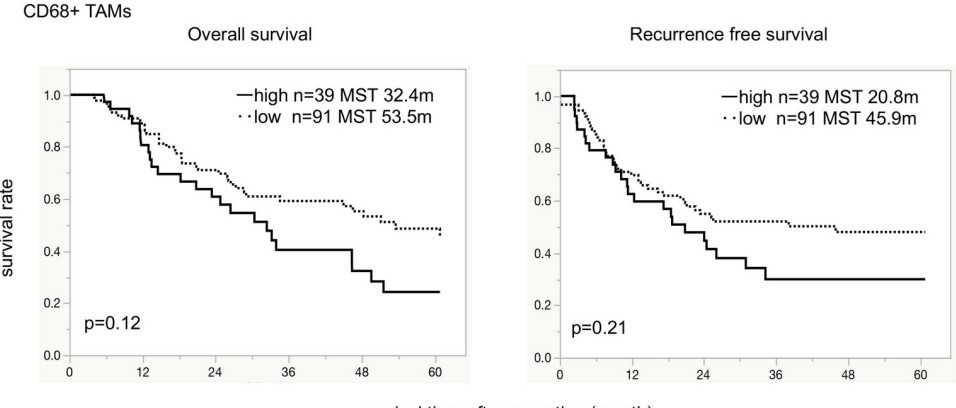

**Fig 3. Overall survival and recurrence-free survival for tumor-associated macrophages.** Kaplan-Meier survival curve indicates that there is no significant difference in OS and RFS between the high- and the low-infiltrating CD68+ TAMs groups. TAMs: tumor-associated macrophages, OS: overall survival, RFS: recurrence-free survival, MST: median survival time.

**Table 4. Univariate and multivariate cox regression analysis for overall survival in 130 patients with BTC.**

| | | Univariate | | | Multivariate | | |
|---|---|---|---|---|---|---|---|
| | | HR | 95% CI | P value | HR | 95% CI | P value |
| T category | pT≧3 | 1.76 | 1.06–2.91 | *0.028 | 1.36 | 0.76–2.42 | 0.3 |
| Lymph node metastasis | | 2.71 | 1.63–4.49 | *<0.001 | 2.0 | 1.08–3.66 | *0.02 |
| Distant metastasis | | 5.31 | 2.37–10.7 | *<0.001 | 3.34 | 1.42–7.32 | *0.007 |
| Serum CEA level | ≧5 ng/ml | 1.62 | 0.83–2.93 | 0.14 | | | |
| Serum CA19-9 level | ≧37 U/ml | 2.0 | 1.17–3.48 | *0.01 | 1.72 | 1.00–3.01 | *0.04 |
| CD8 TILs | High | 0.72 | 0.44–1.19 | 0.2 | | | |
| CD4 TILs | High | 0.56 | 0.34–0.95 | *0.03 | 0.66 | 0.37–1.18 | 0.16 |
| FOXP3 TILs | High | 0.63 | 0.35–1.08 | 0.1 | | | |
| CD68 TAMs | High | 1.66 | 0.96–2.76 | 0.05 | | | |

*p < 0.05

BTC: biliary tract cancers, CEA; carcinoembryonic antigen, CA19-9; carbohydrate antigen 19–9, TILs; tumor infiltrating lymphocytes, TAMs; tumor associated macrophages, HR; Hazards ration, CI; confidence interval.

## Association between Immunoscore and survival outcomes

To understand the influence of immune cells infiltration into TME on tumor progression, we evaluated the Immunoscore based on the status of infiltrating immune cells. The number of cases was 18 for score 4, 41 for score 3, 40 for score 2, 29 for score 1, and 2 for score 0. A total of 59 cases had high Immunoscore (3–4) and 71 cases had low Immunoscore (0–2). Patients with high Immunoscore showed significantly superior OS and RFS than those with low Immunoscore (median OS 60.8 vs. 26.4 months, respectively, *p = 0.001*; median RFS not reached vs. 17.2 months, respectively, *p < 0.001*; Fig 4).

## Multivariate analysis of Immunoscore

Multivariate analysis of the patients was performed with clinicopathological predictors and Immunoscore (Table 2). For OS, low Immunoscore, positive lymph node metastasis, presence of distant metastasis, and high serum CA19-9 level were independent poor prognostic factors

**Table 5. Univariate and multivariate cox regression analysis for recurrence free survival in 130 patients with BTC.**

| | | Univariate | | | Multivariate | | |
|---|---|---|---|---|---|---|---|
| | | HR | 95% CI | | HR | 95% CI | P value |
| T category | pT≧3 | 2.36 | 1.44–3.89 | *<0.001 | 2.20 | 1.22–3.99 | *0.008 |
| Lymph node metastasis | | 2.99 | 1.82–4.91 | *<0.001 | 1.85 | 0.99–3.4 | 0.05 |
| Distant metastasis | | 5.80 | 2.61–11.6 | *<0.001 | 3.1 | 1.25–7.18 | *0.01 |
| Serum CEA level | ≧5 ng/ml | 2.16 | 1.12–3.91 | *0.02 | 1.72 | 0.85–3.27 | 0.13 |
| Serum CA19-9 level | ≧37 U/ml | 1.99 | 1.19–3.37 | *0.008 | 1.5 | 0.88–2.59 | 0.13 |
| CD8 TILs | High | 0.68 | 0.42–1.11 | 0.12 | | | |
| CD4 TILs | High | 0.43 | 0.27–0.72 | *0.001 | 0.57 | 0.33–1.03 | 0.06 |
| FOXP3 TILs | High | 0.57 | 0.33–0.97 | *0.03 | 0.73 | 0.41–1.27 | 0.27 |
| CD68 TAMs | High | 1.47 | 0.88–2.42 | 0.13 | | | |

*p < 0.05

BTC: biliary tract cancers, CEA; carcinoembryonic antigen, CA19-9; carbohydrate antigen 19–9, TILs; tumor infiltrating lymphocytes, TAMs; tumor associated macrophages, HR; Hazards ration, CI; confidence interval.

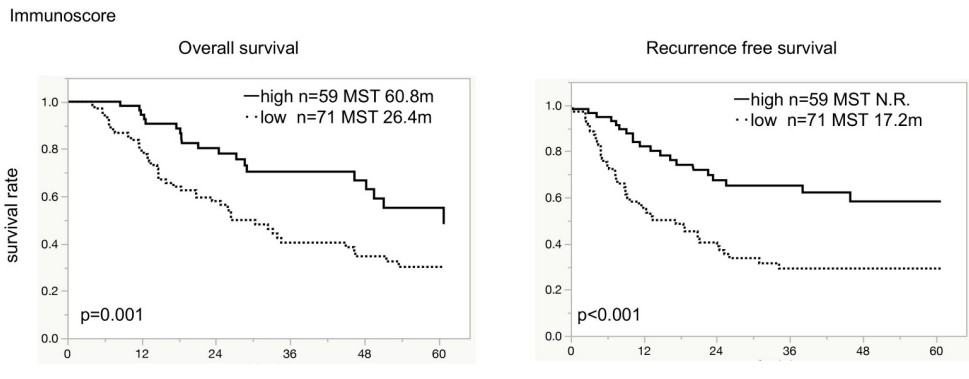

**Fig 4. Overall survival and recurrence-free survival for tumor-infiltrating immune cells score.** Immunoscore was assigned a number from 0 to 4, according to CD8+ high, CD4+ high, FOXP3+ high, and CD68+ low-infiltration. Kaplan-Meier survival curve indicates that the high Immunoscore group showed significantly superior OS and RFS than the low Immunoscore group. OS: overall survival, RFS: recurrence-free survival, MST: median survival time, N. R.: not reached.

(hazards ratio 2.05, 3.48, 1.7, and 2.05, respectively; *p = 0.01, p = 0.005, p = 0.05*, and *p = 0.01*, respectively). For RFS, low Immunoscore, T classification of pT3-4, and presence of distant metastasis were independent poor prognostic factors with a hazards ratio of 2.41 (*p = 0.001*), 2.16 (*p = 0.005*), and 3.58 (*p = 0.005*), respectively (Table 6).

## Discussion

This study indicated that although TILs, including CD8+ T cells, CD4+ T cells, and FOXP3+ T cells, did not correlate with clinicopathological factors and each T cell alone was not an independent prognostic factor, TILs tended to improve the prognosis of patients with BTC. On the other hand, high-infiltration with macrophages did not show a poor prognosis, and it is also not a significant independent prognostic factor for OS and RFS. When every infiltrating T cell and macrophage in TME were comprehensively scored, a high Immunoscore group had significantly longer OS and RFS and was an independent prognostic factor in the multivariate analysis. Our findings indicated that in BTC, evaluating TILs and TAMs in TME comprehensively was better than evaluating one TIIC, such as T cell or macrophage, each by each.

TILs are composed of T cells, B cells, and natural killer (NK) cells [21]. Among them, T cells play the most central role in TME. In the current study, we evaluated CD8+ T cell, CD4 + T cell, and regulatory T cells. CD8+ T cells have been reported to have several subsets such

**Table 6. Multivariate cox regression analysis for overall and recurrence free survival in 130 patients with BTC.**

|  |  | Multivariate for OS | | | Multivariate for RFS | | |
|---|---|---|---|---|---|---|---|
|  |  | HR | 95% CI | P value | HR | 95% CI | P value |
| T category | pT ≧ 3 | 1.35 | 0.76–2.39 | 0.28 | 2.16 | 1.26–3.76 | *0.005 |
| Lymph node metastasis |  | 2.07 | 1.11–3.78 | *0.02 | 1.78 | 0.97–3.21 | 0.06 |
| Distant metastasis |  | 3.48 | 1.47–7.68 | *0.005 | 3.58 | 1.49–8.01 | *0.005 |
| Serum CA19-9 level | ≧37 U/ml | 1.7 | 0.99–2.97 | *0.05 | 1.55 | 0.92–2.63 | 0.09 |
| Immunoscore | ≧3 | 2.05 | 1.18–3.67 | *0.01 | 2.41 | 1.41–4.23 | *0.001 |

*p < 0.05

BTC: biliary tract cancers, CA19-9; carbohydrate antigen 19–9, OS; overall survival, RFS; recurrence free survival, HR; Hazards ration, CI; confidence interval.

as naïve CD8 T cell, memory CD8 T cell, and effector CD8 T cell [22]. The effector CD8 T cells are well-known and play a cytotoxic role to attack tumors directly [23]. Many researchers revealed a correlation between infiltration of CD8+ TILs and survival outcomes [24]. Moreover, CD8+ TILs can also be useful for predicting the effects of immunotherapy [25, 26]. We also reported an association between CD8+ infiltration and patient outcome in BTC [20]. Unlike CD8+ TILs, CD4+ TILs might contribute to anti-tumor immunity via cytokines [27, 28]. However, the function of CD4+ TILs in TME is still unclear due to the existence of many subsets [29]. In the current study, CD4+ TILs helped anti-tumor immunity. Regulatory T cells (Tregs) were initially characterized as CD4+/CD25+ T cells, and Tregs cell markers are known as FOXP3 [30]. FOXP3+ T cell might suppress the activity of cytotoxic T cells via cytokines; therefore, high-infiltration with FOXP3+ TILs is correlated with poor prognosis in several cancers [31, 32]. However, some researchers indicated that FOXP3 is one of the unfavorable prognostic factors in colorectal cancer [33]. The reason of these discrepancies is that the role of immune cells in the microenvironment differs depending on the origin of the tumor [34]. FOXP3+ TILs suppress tumor-promoting inflammatory responses under the presence of the enteric bacteria [33]. That's why high-infiltration with FOXP3+ is not always associated the good prognostic factor, and similar reacts may occur in BTC. There are many previous reports about TILs, however, in TME, each immune cell interacts with the other. We considered that it is necessary to evaluate immune cells comprehensively as TME.

Basically, TAMs are involved in tumor progression. However, in a meta-analysis, high-infiltrating TAMs were associated with a poor prognosis in gastric, breast, and ovarian cancer whereas these were associated with a good prognosis in colorectal cancer [9]. The reason for this discrepancy is that macrophages have two main phenotypes, M1 and M2. M1 macrophages induced by cytokines, such as transforming growth factor (TGF)-β, interleukin (IL)-6, and IL-10, have anti-tumor activity. On the other hand, M2 macrophages induced by IL-4 and IL-13 play a key role in tumor progression and metastasis [35]. CD68 is known as a pan-macrophage marker and is correlated with a poor prognosis in breast cancer and lymphoma [36–38]. In this study, CD68 was used as a marker for TAMs instead of CD80 for M1, CD163 or CD206 for M2 specific macrophage marker [39]. Therefore, high-infiltration with CD68 + TAMs was not correlated with tumor progression. To evaluate tumor progression with each specific marker, it might be useful to investigate the effect of macrophages on patient outcomes. Furthermore, although the infiltration of CD68+ TAMs did not show correlation with the infiltration of TILs in our current study, it has been reported that TAMs suppress immune reaction to tumors in TME [40]. Further evaluation of immunosuppression markers with iNOS or IDO may help understand the function of immune cells in TME.

We investigated whether it is possible to comprehensively assess T-cell infiltration using CD3 marker as a pan-T cell marker. Cases with the number of infiltrating CD3+ TILs that varied from 0 to 480/high power field (cut-off 52) (S4A Fig) and 52.3% (68 out of 130 cases) were assigned to the high-infiltration CD3+ TILs group (S4B and S4C Fig). The high-infiltration CD3+ TILs group showed significantly superior RFS than the low-infiltration CD3+ TILs group. For OS, there is no significant difference between the high- and low- infiltration CD3 + TILs groups (median OS 46.4 vs. 45 months, respectively, *p = 0.08*; median RFS 34.2 vs. 17.3 months, respectively, *p = 0.03*) (S4D Fig). Comprehensive evaluation with CD3+ TILs and CD68+ TAMs showed the low-infiltration CD3+ TILs and high-infiltration CD68+ TAM group was an independent poor prognostic factor in only OS (others vs. CD3+ low and CD68 + high; median OS 48.3 vs 20.8 months, *p = 0.01*; hazards ratio 2.65, *p = 0.01*), but not in RFS (median RFS 26.1 vs. 11.7 months, *p = 0.09*; hazards ratio 1.54, *p = 0.24*) (S4E Fig, S2 Table). Immunoscore, which is a score to evaluate each immune cell infiltration, was more associated with OS and RFS. This indicates that evaluating various subsets of immune cells in TME

would be a better prediction factor of OS and RFS. However, evaluating all subsets by immunohistochemical staining with each marker are complicated and may not be feasible. Therefore, to apply the results obtained in our study to clinical in the future, more convenient and simple evaluation method will be necessary, such as evaluating immune cell infiltration with only H & E staining [41].

Not only malignancy of the tumor itself but also tumor immunity in TME, including TILs and TAMs, B cells, NK cells, neutrophils, and dendric cells, is involved in tumor progression and patient outcomes [42]. Recently, various types of therapeutic strategies targeting immune cells have been developed, including adoptive cell therapy, TIL therapy, T cell receptor gene therapy, chimeric antigen receptor (CAR) T cell therapy, NK cell therapy, and CAR NK cell therapy [43–45]. Nowadays, several clinical trials on adoptive T cell therapy for BTC are ongoing [46, 47]. We believe that our findings contribute to the development and selection of a treatment strategy for BTC.

There are some limitations to this study. First, this study was a retrospective single-center cohort. Retrospective nature limits our understanding of the associations while single-center nature limits the generalizability of the findings. Second, there is not validation cohorts in this study to check the cutoff value are appropriate or not. Third, it included only five subtypes of BTCs, i.e., intrahepatic, perihilar, and distal bile duct cancer, gallbladder cancer, and ampullary cancer. This limits the application of findings to other cancer types. Fourth, we did not analyze other immune cells by immunohistochemical staining, such as B cells, neutrophils, or dendric cells. These immune cells may also affect tumor progression in TME. Lastly, we did not clarify the polarization of the macrophages. CD68 was initially proposed to exclude macrophages, however, it has recently been reported to be expressed in dendritic cells, tumor cells, endothelial cells, and fibroblasts.

In conclusion, we found that the high Immunoscore group had significantly longer OS and RFS and was an independent prognostic factor. Our findings indicated that in biliary tract cancer, the evaluation of infiltrating immune cells in TME was useful to predict patient prognosis.

## Supporting information

**S1 Fig. H & E staining of tissue micro array.**
(TIF)

**S2 Fig. The number of infiltrating immune cells.** HPF; high power field.
(TIF)

**S3 Fig. Receiver operating characteristic (ROC) curve analyses for 5-year RFS.** TILs; tumor-infiltrating lymphocytes, TAMs; tumor associated macrophages. AUC; area under the curve.
(TIF)

**S4 Fig. The impact of CD3+ TILs infiltration.** a: The number of infiltrating CD3+ TILs. b, c: IHC staining with CD3 antibody. b: low infiltration, c: high infiltration. d: overall survival and recurrence-free survival for CD3+ TILs. e: overall survival and recurrence-free survival for CD3+ TILs and CD68+ TAMs. HPF; high power field, IHC; immunohistochemical staining, TILs; tumor-infiltrating lymphocytes, TAMs; tumor associated macrophages.
(TIF)

**S1 Table. Multivariate cox regression analysis for overall and recurrence free survival in 130 patients with BTC.**
(DOCX)

**S2 Table. Multivariate cox regression analysis for overall and recurrence free survival in 130 patients with BTC.**
(DOCX)

## Acknowledgments

We would like to thank Editage (www.editage.com) for English language editing.

## Author Contributions

**Conceptualization:** Ryota Tanaka, Kenjiro Kimura.

**Data curation:** Ryota Tanaka, Kenjiro Kimura.

**Investigation:** Kenjiro Kimura.

**Writing – original draft:** Ryota Tanaka, Kenjiro Kimura.

**Writing – review & editing:** Shimpei Eguchi, Go Ohira, Shogo Tanaka, Ryosuke Amano, Hiroaki Tanaka, Masakazu Yashiro, Masaichi Ohira, Shoji Kubo.

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
