## [Decision Letter · Decision Letter 0]

18 Aug 2022

PONE-D-22-22499Tumor-Infiltrating Lymphocytes and Macrophages as a Significant Prognostic Factor in Biliary Tract CancerPLOS ONE

Dear Dr. Kenjiro Kimura,

Thank you for submitting your manuscript to PLOS ONE. After careful consideration, we feel that it has merit but does not fully meet PLOS ONE’s publication criteria as it currently stands. Therefore, we invite you to submit a revised version of the manuscript that addresses the points raised during the review process.  The study has merit.

Please submit your revised manuscript within 60 days. If you will need more time than this to complete your revisions, please reply to this message or contact the journal office at plosone@plos.org. Please include the following items when submitting your revised manuscript:A rebuttal letter that responds to each point raised by the academic editor and reviewer(s). You should upload this letter as a separate file labeled 'Response to Reviewers'.A marked-up copy of your manuscript that highlights changes made to the original version. You should upload this as a separate file labeled 'Revised Manuscript with Track Changes'.An unmarked version of your revised paper without tracked changes. You should upload this as a separate file labeled 'Manuscript'.

We look forward to receiving your revised manuscript.

Kind regards,

Gianfranco D. Alpini

Academic Editor

PLOS ONE

Journal Requirements:

" ext-link-type="uri" xlink:type="simple">https://journals.plos.org/plosone/s/file?id=ba62/PLOSOne_formatting_sample_title_authors_affiliations.pdf"

2. Our staff editors have determined that your manuscript is likely within the scope of our Early Detection, Screening and Diagnosis of Cancer Call for Papers. This editorial initiative is headed by in-house PLOS editors. This Call for Papers aims to explore recent advances in the early detection of cancer and implications of these advances for patient survival. Additional information can be found on our announcement page: https://collections.plos.org/call-for-papers/early-detection-screening-and-diagnosis-of-cancer/

If you would like your manuscript to be considered for this collection, please let us know in your cover letter and we will ensure that your paper is treated as if you were responding to this call.  Please note that being considered for the Call for Papers does not require additional peer review beyond the journal’s standard process and will not delay the publication of your manuscript if it is accepted by PLOS ONE. If you would prefer to remove your manuscript from collection consideration, please specify this in the cover letter.

Reviewers' comments:

Reviewer's Responses to Questions

**Comments to the Author**

1. Is the manuscript technically sound, and do the data support the conclusions?

Reviewer #1: Partly

Reviewer #2: Partly

2. Has the statistical analysis been performed appropriately and rigorously? 

Reviewer #1: Yes

Reviewer #2: Yes

3. Have the authors made all data underlying the findings in their manuscript fully available?

Reviewer #1: Yes

Reviewer #2: Yes

4. Is the manuscript presented in an intelligible fashion and written in standard English?

Reviewer #1: Yes

Reviewer #2: Yes

5. Review Comments to the Author

Reviewer #1: Tumor-Infiltrating Lymphocytes and Macrophages as a Significant Prognostic Factor in Biliary Tract Cancer

In this manuscript the authors study the impact of tumor-infiltrating lymphocytes (TILs) and tumor-associated macrophages (TAMs) on the prognosis of biliary tract cancer (BTC), concepts that are not fully elucidated and understood. They study the effects of the various immune cells infiltration in tumor microenvironment (TME). For their in vivo experiments they used 130 patients with BTC who underwent surgical treatment and evaluated TILs and TAMs with immunohistochemical staining. In their results they show CD8-high, CD4-high, FOXP3-high, and CD68-low in TME as one factor, and they we calculated the immune score according to the number of factors. The high immune-score group showed significantly superior overall survival (OS) and recurrence-free survival (RFS) and the low immune-score group (median OS, 60.8 vs. 26.4 months, p = 0.001; median RFS not reached vs. 17.2 months, p 0.001). Furthermore, high immune-score was an independent good prognostic factor for OS and RFS (hazards ratio 2.05 and 2.41 and p = 0.01 and p = 0.001, respectively). Finally, they conclude that high immune-score group had significantly superior OS and RFS and was an independent good prognostic factor for OS and RFS. Despite the interesting points elucidated by the authors in this study, before proceeding with publication, the following questions must be addressed by the authors:

1. In the Figure 1, the authors shown representative immunohistochemistry images of high and low infiltration of CD8-high, CD4-high, and FOXP3-high TILs and CD68-low CD8+, CD4+, FOXP3+, and CD68+ cells, in human patients. First, the authors need to mention the kind of microscope by which these pictures are taken. Also, they please a graph shown a numerical evaluation of the staining it should be appreciable by the readers. Furthermore, need to add some arrows to show the infiltration of the above cells in the tissue. In the Figure1, the authors consider two groups of patients with low and high infiltration, it should be interesting if in the Figure 1 they add also images of the healthy control group.

2. In the Table1, the authors listed some of the clinicopathological characteristics of the 130 patients with BTC. Please, the authors must complete the list, give some more info such as age, eventually pharmacological treatment of the patients, stage of the illness ecc.

3. In the tissue microarray construction paragraph, the authors ensured that representative tumor cell-rich areas are HE stained with a light microscope and were sent to create TMA blocks. The authors must include some HE images of the tumor sections in their results. So the readers can appreciate the percentage of TILs on HE stained samples.

4. Is well known that immunohistochemical markers used to identify M1 and M2 TAMs are the keystones of TAM evaluation. In this manuscript the authors study the tumor-associated macrophages (TAMs) in BTC. It should be interesting if the authors extent their experiments (immunohistochemistry or PCR or blots) in other markers that characterized the TAMs, such as, CD11c, CD86, iNOS, pSTAT1 (M1 markers), CD163, CD204, CD206 (M2 markers) or at least some of them.

Reviewer #2: This is a very interesting study looking at the correlation between TIL in tumors from Biliary Tract cancer.

However, the authors only look at high and low levels of infiltrating immune cells, control/normal tissue analysis is needed.

You may need to assess the co-staining of CD4+ and Foxp3 staining, since there may be overlap, to conclusively determine specific cell type, as well as CD3+ with CD8+/CD4+ cells (instead of them each individually).

Images in Figure 2 are hard to visualize. It may be necessary to use Immunofluorescence to visualize co-staining.

In the discussion, the authors state that role of CD8+ T cells as a cytotoxic role. This is not necessarily true. There are numerous subsets of CD8+ T cell and the authors did not access these.

Although specific patient data may be restricted, the authors did not access potential cofounders that could alter results from this study in the discussion.

6. PLOS authors have the option to publish the peer review history of their article (what does this mean?). If published, this will include your full peer review and any attached files.

Reviewer #1: No

Reviewer #2: No

---

## [Author Response · Author response to Decision Letter 0]

4 Dec 2022

Point by point response to Reviewer 1 Comments

We are grateful for your comments which have helped to improve our manuscript. As indicated in the responses that follow, we have taken your comments into account in our manuscript. 

1. In the Figure 1, the authors shown representative immunohistochemistry images of high and low infiltration of CD8-high, CD4-high, and FOXP3-high TILs and CD68-low CD8+, CD4+, FOXP3+, and CD68+ cells, in human patients. First, the authors need to mention the kind of microscope by which these pictures are taken. Also, they please a graph shown a numerical evaluation of the staining it should be appreciable by the readers. Furthermore, need to add some arrows to show the infiltration of the above cells in the tissue. In the Figure1, the authors consider two groups of patients with low and high infiltration, it should be interesting if in the Figure 1 they add also images of the healthy control group.

Response: We added the information of microscope in page 9, line 92-93. We added the figure of the number of infiltrating immune cells as Supplementary Figure 2. 

We added arrows to the stained immune cells in Figure 1.

In this current study, we evaluated the infiltration of immune cells into the tumor microenvironment in patients with biliary tract cancer. Also, we investigated how well the infiltration of immune cells can predict clinical prognosis. Because healthy patients do not have tumor, it is impossible to evaluate immune cell infiltrations into their tumor microenvironment. In addition, it is difficult to obtain biliary tract specimens from healthy non-cancer-bearing patients.

2. In the Table1, the authors listed some of the clinicopathological characteristics of the 130 patients with BTC. Please, the authors must complete the list, give some more info such as age, eventually pharmacological treatment of the patients, stage of the illness ecc.

Response: We already listed age and stage in Table 1. We added the information of pharmacological treatment into Table 1. 

3. In the tissue microarray construction paragraph, the authors ensured that representative tumor cell-rich areas are HE stained with a light microscope and were sent to create TMA blocks. The authors must include some HE images of the tumor sections in their results. So the readers can appreciate the percentage of TILs on HE stained samples. 

Response: We added the images of HE staining as Supplementary Figure 1.

4. Is well known that immunohistochemical markers used to identify M1 and M2 TAMs are the keystones of TAM evaluation. In this manuscript the authors study the tumor-associated macrophages (TAMs) in BTC. It should be interesting if the authors extent their experiments (immunohistochemistry or PCR or blots) in other markers that characterized the TAMs, such as, CD11c, CD86, iNOS, pSTAT1 (M1 markers), CD163, CD204, CD206 (M2 markers) or at least some of them.

Response: I totally agree with your point. Performing double staining using various cell surface markers will allow us to understand immune cell infiltration in TME in more detail. However, our future goal is to examine how to apply this result to clinical practice, and we would like to replace it with a simple method. We evaluated CD3 positive cells as a pan-T cell marker. We found that CD3 is useful to understand the infiltration of immune cells in TME, although the accuracy decreased compared to each immune cell evaluation. In the future, we will investigate whether predicting prognosis is possible by evaluating immune cell infiltration with H E staining. We added the paragraph in page 31-32, line 305-325. Also, we added Supplementary Figure 4 and Supplementary Table 2.

Point by point response to Reviewer 2 Comments

We are grateful for your comments which have helped us greatly improve our manuscript. As indicated in the responses that follow, we have taken your comments into account in our manuscript. 

1. However, the authors only look at high and low levels of infiltrating immune cells, control/normal tissue analysis is needed.

Response: In this current study, we evaluated the infiltration of immune cells into the tumor microenvironment in patients with biliary tract cancer. Also, we investigated how well the infiltration of immune cells can predict clinical prognosis. Because healthy patients do not have tumor, it is impossible to evaluate immune cell infiltrations into their tumor microenvironment. In addition, it is difficult to obtain biliary tract specimens from healthy non-cancer-bearing patients.

2. You may need to assess the co-staining of CD4+ and Foxp3 staining, since there may be overlap, to conclusively determine specific cell type, as well as CD3+ with CD8+/CD4+ cells (instead of them each individually).　

Response: I totally agree with your point. Performing double staining using various cell surface markers will allow us to understand immune cell infiltration in TME in more detail. However, our future goal is to examine how to apply this result to clinical practice, and we would like to replace it with a simple method. We evaluated CD3 positive cells as a pan-T cell marker. We found that CD3 is useful to understand the infiltration of immune cells in TME, although the accuracy decreased compared to each immune cell evaluation. In the future, we will investigate whether predicting prognosis is possible by evaluating immune cell infiltration with H E staining. We added the paragraph in page 31-32, line 305-325. Also, we added Supplementary Figure 4 and Supplementary Table 2.

3. Images in Figure 2 are hard to visualize. It may be necessary to use Immunofluorescence to visualize co-staining.

Response: We added arrows to the stained immune cells in Figure 1. Also, we changed the contrast and magnification to improve the clarity of visualization. 

4. In the discussion, the authors state that role of CD8+ T cells as a cytotoxic role. This is not necessarily true. There are numerous subsets of CD8+ T cell and the authors did not access these.

Response: We added the description of CD8 subsets in page 28, lines 264-267.

---

## [Editor Report · Decision Letter 1]

27 Dec 2022

Tumor-Infiltrating Lymphocytes and Macrophages as a Significant Prognostic Factor in Biliary Tract Cancer

PONE-D-22-22499R1

Dear Dr. Kenjiro Kimura,

We’re pleased to inform you that your manuscript has been judged scientifically suitable for publication and will be formally accepted for publication once it meets all outstanding technical requirements.

Kind regards,

Gianfranco D. Alpini

Academic Editor

PLOS ONE
---

## [Editor Report · Acceptance letter]

4 Jan 2023

PONE-D-22-22499R1 

Tumor-Infiltrating Lymphocytes and Macrophages as a Significant Prognostic Factor in Biliary Tract Cancer 

Dear Dr. Kimura:

I'm pleased to inform you that your manuscript has been deemed suitable for publication in PLOS ONE. Congratulations! Your manuscript is now with our production department. 

Kind regards, 

on behalf of

Dr. Gianfranco D. Alpini 

Academic Editor

PLOS ONE